# Diagnostic accuracy of the WHO clinical definitions for dengue and implications for surveillance: A systematic review and meta-analysis

**Nader Raafat**[1,2], **Shanghavie Loganathan**[2], **Mavuto Mukaka**[1,3], **Stuart D. Blacksell**[1,3], **Richard James Maude**[1,3,4]*

1 Mahidol-Oxford Tropical Medicine Research Unit, Faculty of Tropical Medicine, Mahidol University, Bangkok, Thailand, 2 Oxford Medical School, University of Oxford, Oxford, United Kingdom, 3 Centre for Tropical Medicine and Global Health, Nuffield Department of Medicine, University of Oxford, Oxford, United Kingdom, 4 Harvard TH Chan School of Public Health, Harvard University, Boston, Massachusetts, United States of America

* richard@tropmedres.ac

**Data Availability Statement:** All relevant data are within the manuscript and its Supporting Information files.

## Abstract

### Background

Dengue is the world's most common mosquito-borne virus but remains diagnostically challenging due to its nonspecific presentation. Access to laboratory confirmation is limited and thus most reported figures are based on clinical diagnosis alone, the accuracy of which is uncertain. This systematic review assesses the diagnostic accuracy of the traditional (1997) and revised (2009) WHO clinical case definitions for dengue fever, the basis for most national guidelines.

### Methodology/Principal findings

PubMed, EMBASE, Scopus, OpenGrey, and the annual Dengue Bulletin were searched for studies assessing the diagnostic accuracy of the unmodified clinical criteria. Two reviewers (NR/SL) independently assessed eligibility, extracted data, and evaluated risk of bias using a modified QUADAS-2. Additional records were found by citation network analysis. A meta-analysis was done using a bivariate mixed-effects regression model. Studies that modified criteria were analysed separately. This systematic review protocol was registered on PROSPERO (CRD42020165998). We identified 11 and 12 datasets assessing the 1997 and 2009 definition, respectively, and 6 using modified criteria. Sensitivity was 93% (95% CI: 77–98) and 93% (95% CI: 86–96) for the 1997 and 2009 definitions, respectively. Specificity was 29% (95% CI: 8–65) and 31% (95% CI: 18–48) for the 1997 and 2009 definitions, respectively. Diagnostic performance suffered at the extremes of age. No modification significantly improved accuracy.

### Conclusions/Significance

Diagnostic accuracy of clinical criteria is poor, with significant implications for surveillance and public health responses for dengue control. As the basis for most reported figures, this

**Funding:** Mahidol-Oxford Tropical Medicine Research Unit (MORU) is funded by the Wellcome Trust of Great Britain [106698/Z/14/Z, https://wellcome.org/]. RJM is supported by a grant from the Research Council of Norway [285188, https://www.forskningsradet.no/en/]. The funders had no role in study design, data collection and analysis, decision to publish, or preparation of the manuscript.

**Competing interests:** The authors have declared that no competing interests exist.

has relevance to policymakers planning resource allocation and researchers modelling transmission, particularly during COVID-19.

## Author summary

Dengue is the most common mosquito-borne disease worldwide, with half the world's population living in at-risk areas, yet it remains difficult to diagnose. Existing laboratory tests have highly variable performance, and access to them remains limited in most dengue-endemic regions. Thus, most dengue cases are diagnosed on clinical criteria alone. While national guidelines vary, most are based on the WHO case definitions, produced in 1997 and revised in 2009. Here, we assess the diagnostic accuracy of both definitions and find that they have good sensitivity but poor specificity, particularly problematic given the co-circulation of multiple febrile illnesses in these regions. This makes it difficult for policymakers and researchers to model transmission, assess the introduction of new pathogens to a region, and correctly prioritise control measures and vaccination programmes in a region-specific manner. This is exacerbated by the ongoing COVID-19 pandemic, given rising cases of both diseases and the stark difference in necessary control measures. As such, improvements in dengue diagnostic and reporting practice are increasingly urgent. This could be achieved by incorporating symptom absence into clinical criteria, weighting symptoms depending on strength of association with dengue or timing within disease course, or using clinical criteria to allocate limited testing resources in borderline cases.

## Introduction

Dengue is the most common mosquito-borne virus worldwide, with an estimated 390 million annual infections globally (last calculated in 2010) [1]. Although the majority of infections are asymptomatic, they likely contribute to viral transmission [1], similar to the ongoing COVID-19 pandemic. As healthcare systems deal with COVID-19, many countries in Latin America and Asia are reporting an increase in dengue cases [2,3], raising concerns of a 'double epidemic' that could overwhelm fragile health systems. As clearly evidenced by COVID-19, the global importance of local disease control cannot be overstated, and it is therefore essential that the current pandemic does not lead to setbacks in dengue control [4]. However, that is only possible if accurate transmission data are available, which is not the case for dengue.

Dengue lacks the robust standardisation of WHO reporting found in other infections such as malaria. Aside from high levels of underreporting [5], the diagnostic accuracy of reported cases remains unclear. Despite recent developments in dengue diagnostics, there is significant variation in accuracy between different tests and different assays of the same test [6]. Access to testing is limited and not mandated in many dengue-endemic countries [7]. Consequently, confirmatory testing is often not done, with only 43% of cases reported to the Pan-American Health Organisation in 2019 being laboratory-confirmed [8]. Equivalent reports could not be found for the Western Pacific and Southeast Asia, although research studies have found low confirmation rates in these regions [9,10]. Thus, most reported cases are likely to be based solely on clinical diagnosis, the accuracy of which has not been formally studied.

Guidelines for the clinical diagnosis of dengue were published by the WHO in 1997 and 2009. The 1997 ('traditional') definition classifies cases into dengue fever (DF), dengue

**Table 1. PICOS statement.**

| Domain | Summary |
|---|---|
| Population | Febrile patients in dengue endemic areas |
| Intervention | Strict use of WHO clinical definitions of dengue (1997 or 2009) |
| Comparison | Confirmatory laboratory tests for dengue |
| Outcome | Sensitivity, specificity, and likelihood ratios of WHO clinical definitions of dengue |
| Study design | Systematic review |

haemorrhagic fever (DHF), and dengue shock syndrome (DSS) [11]; while the 2009 ('revised') definition classifies cases into dengue and severe dengue [12]. In both guidelines, laboratory confirmation is not necessary to diagnose 'probable' dengue in endemic locations.

The development of the WHO case classification has been reviewed elsewhere [13], and whilst methodologically robust, the aim was to improve early prediction of severe disease, rather than distinguish dengue from non-dengue febrile illnesses. Thus, most studies have focused on the guidelines' prognostic value. In this systematic review, we assess the diagnostic performance of the 1997 and 2009 WHO clinical definitions of 'probable dengue' in febrile patients and discuss the implications for surveillance and control.

## Methods

The protocol was registered on PROSPERO on 27/01/2020 (CRD42020165998). The PICOS statement is outlined in Table 1.

### Eligibility criteria for studies

**Study design and participants.** Studies comparing the WHO diagnostic criteria to a suitable reference standard (see below) in patients with unexplained fever were included. There were no limitations on demographics, fever duration, healthcare setting, or geographical region. Studies were excluded if they only recruited confirmed or suspected dengue patients or excluded any dengue serotypes.

**Index test and reference standard.** The index tests were the 1997 [11] and 2009 [12] WHO clinical definitions for dengue. Studies applying either definition without modification (Table 2) were included. Studies that modified the WHO criteria were analysed separately to

**Table 2. Traditional (1997) and revised (2009) WHO clinical definitions of dengue fever.**

| 1997 ('traditional') definition | 2009 ('revised') definition |
|---|---|
| Occurrence at the same location and time as other confirmed cases of dengue fever. Acute febrile illness with two or more of the following: • Headache • Retro-orbital pain • Myalgia • Arthralgia • Rash • Haemorrhagic manifestations: ○ A positive tourniquet test (≥20 petechiae per 2.5cm square) ○ Petechiae, ecchymoses, or purpura ○ Bleeding from the mucosa, gastrointestinal tract, injection sites, or other locations ○ Haematemesis or melaena ○ Leukopenia | Live in/travel to dengue-endemic area. Fever and 2 of the following criteria: • Nausea, vomiting • Rash • Aches and pains • Tourniquet test positive • Leukopenia • Any warning sign: ○ Abdominal pain or tenderness ○ Persistent vomiting ○ Clinical fluid accumulation ○ Mucosal bleed ○ Lethargy, restlessness ○ Liver enlargement >2 cm ○ Increase in haematocrit concurrent with a rapid decrease in platelet count |

determine what effect this had. With no accepted reference standard for dengue, any of the following, as per WHO guidance [12], were acceptable: IgM or IgG serology, plaque reduction neutralisation test or hemagglutination inhibition, NS1 antigen/antibody test, (RT-)PCR, or virus isolation.

## Search methodology

PubMed, EMBASE, Scopus, and OpenGrey were searched using the strings outlined in S1 Table. Records published from 1997 to the last search on 19/1/2020 were included, with no restrictions on type of publication or language.

Search results were pooled and duplicates removed using EndNote X9 (Clarivate Analytics, USA). Abstracts of all articles and short notes in the annual Dengue Bulletin (published by WHO SEARO) from 1997–2014 (last available volume) were also included. Titles and abstracts were independently screened by two reviewers (NR and SL). This was repeated for eligible full-text articles, with the reason for exclusion recorded. Any disagreements were resolved by a third reviewer (RJM). Authors of conference abstracts were contacted to identify related peer-reviewed publications. Finally, all articles citing (from The Web of Knowledge) and cited by (from reference lists) included studies were screened. For articles not available on Web of Knowledge, Google Scholar was used. This was repeated until no more studies were identified.

## Analysis

Risk of bias was assessed by two independent reviewers (NR and SL) using a modified QUADAS-2 tool (S2 Table) [14]. Study information and 2x2 table data (principal summary measure) were extracted by one reviewer and verified by a second reviewer (NR and SL). Any disagreements were resolved by a third reviewer (RJM). Authors were contacted for missing information, and if no response was received within 3 weeks this was repeated. If no response was subsequently received, it was recorded as not specified. Further detail can be found in S1 File.

Meta-analysis for sensitivity, specificity, and likelihood ratios for both definitions was done using the MIDAS statistical package [15] on Stata/IC 14 (College Station, TX, USA). This uses a bivariate mixed-effects regression framework to calculate average sensitivity and specificity. Deeks' funnel plot asymmetry test was used to detect publication bias for both meta-analyses.

A forest plot for sensitivity, specificity, and corresponding 95% confidence intervals was obtained. Only studies using unmodified WHO criteria were included in the meta-analysis. Heterogeneity was assessed using the $I^2$ and Chi-square statistics. A separate analysis was carried out excluding studies at high risk of bias.

## Results

### Search results

The original search identified 1471 records. One additional record was included, identified from previous work but not found by the search as it did not mention WHO criteria in the title or abstract. After duplicates were removed, 1088 records remained. Dengue Bulletin provided 340 additional records; the 2005 and 2006 volumes could not be found online and were not screened.

119 full-text articles were assessed for eligibility, of which 16 were included. Citation analysis identified 5 additional records. In total, 21 records were included in the qualitative analysis, and 15 records using unmodified WHO criteria in the meta-analysis (Fig 1).

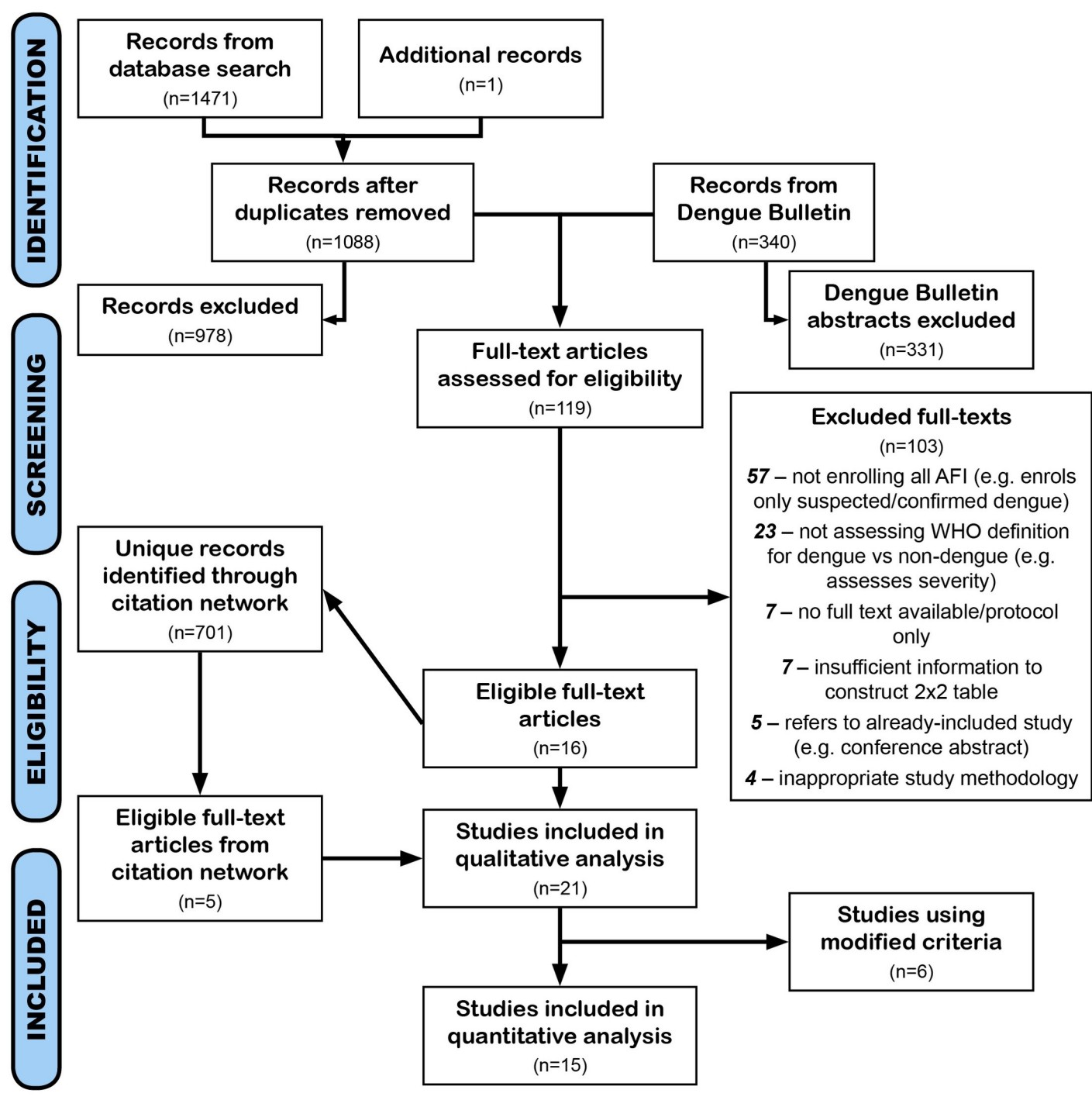

**Fig 1. PRISMA flow diagram for systematic review search results.** AFI: Acute febrile illness. PubMed, EMBASE, Scopus, and OpenGrey were searched for articles assessing the diagnostic accuracy of clinical criteria for dengue diagnosis, Dengue Bulletin articles and short notes were also included. The additional record was identified from previous work but did not mention WHO criteria in the title or abstract. Citation network analysis used Web of Science, Google Scholar, and reference lists. Articles assessing the diagnostic accuracy of unmodified WHO clinical criteria (1997 or 2009) for dengue were included in the meta-analysis, articles using modified criteria were included in qualitative analysis only.

## Study characteristics

Study characteristics and patient flow are summarised in Tables 3 and 4, respectively. Three records were conference abstracts, the remaining 18 came from peer-reviewed journals. Two records [16,17] presented findings from independent studies in the same publication and were thereafter treated as separate, so that final analysis contained 23 separate datasets. 5 out of 23 studies were retrospective. 11 studies were in Asia, 6 South America, 3 Europe (returning travellers), 2 Central America, and 1 Africa. Overall, there were 11 datasets comprising 10,355 patients assessing the traditional (1997) definition, and 12 datasets comprising 9,421 patients assessing the revised (2009) definition; with 6 assessing both definitions. The most common modification to WHO definitions was not using the tourniquet test, in 3 out of 6 modified studies.

## Risk of bias

Risk of bias analysis for included studies is presented in Fig 2. The most common methodological flaw, in 12 out of 23 studies, was the use of an unreliable reference standard (e.g. unpaired IgM serology). The anticipated impact of the bias on each study's estimated sensitivity and specificity, along with the rationale for this choice, is provided in S3 Table.

## Meta-analysis

There was no evidence of publication bias (S1 and S2 Figs). The findings for the 1997 definition are summarised in Fig 3 and S4 Table. Overall sensitivity was 93% (**95% CI:** 77–98, **range:** 13–100), and the specificity was 29% (**95% CI:** 8–65, **range:** 1–99). Positive and negative likelihood ratios were 1.3 (**95% CI:** 0.9–1.9) and 0.24 (**95% CI:** 0.12–0.50) respectively. When studies at high risk of bias were excluded, sensitivity and specificity were 94% (**95% CI:** 83–98, **range:** 13–100) and 27% (**95% CI:** 7–65, **range:** 1–99) respectively.

The findings for the 2009 definition are summarised in Fig 4 and S5 Table. The overall sensitivity was 93% (**95% CI:** 86–96, **range:** 71–99), and the specificity was 31% (**95% CI:** 18–48, **range:** 3–74). Positive and negative likelihood ratios were 1.3 (**95% CI:** 1.1–1.7) and 0.24 (**95% CI:** 0.13–0.45) respectively. When studies at high risk of bias were excluded, sensitivity and specificity were 96% (**95% CI:** 89–98, **range:** 81–99) and 16% (**95% CI:** 7–33, **range:** 3–55) respectively.

Overall, our meta-analysis gave similar results for the two definitions' diagnostic accuracy, with high sensitivity and low specificity for both. This echoes studies that assessed both definitions, with two finding no difference [20,26], three finding a higher sensitivity and lower specificity in the 2009 definition [16,17,33], and two finding the opposite [16,34]. However, there was significant heterogeneity between studies for both definitions (Figs 3 and 4), as reflected in the wide range of reported values, high $I^2$ (97–100%), and statistically significant Chi-squared tests (p<0.0001), even when high-risk studies were excluded.

## Modified criteria

Results from studies using modified criteria are shown in Tables 5 and S6. Diagnostic accuracy for all modifications was similar to the corresponding WHO case definition. Studies that improved [35] or worsened [32,34] both sensitivity and specificity showed high risk of bias in more than one domain (Fig 2) and should be interpreted with caution. Removing the tourniquet test reduced specificity in two studies [33,34], consistent with its association with dengue (see below), although it increased specificity in another study [36].

**Table 3. Characteristics of studies included in the systematic review.**

| Study | Data collection period | Location | Definition assessed | Reference standard(s) |
|---|---|---|---|---|
| Sawasdivorn 2001 [18] | September 1998 – September 1999 | Paediatrics Department, Sawanpracharak Medical Centre, Bangkok, Thailand | 1997 | **ELISA positive**—Armed Forces Research Institute of Medical Sciences ELISA, IgM/IgG not specified, seroconversion criteria not specified. **PCR positive** |
| Martinez 2005 [19] | April 2003 – January 2004 | Health centres in the metropolitan area of Bucaramanga, Colombia | 1997 | **IgM seroconversion or a four-fold rise in titres** (Center for Research in Tropical Diseases of the Industrial University of Santander ELISA) **Virus isolation**—C6/36 mosquito cells |
| Gan 2011 [20][a] | Not specified | Communicable Disease Centre, Singapore | 1997 and 2009 | **PCR positive** **NS1 positive** **IgM seroconversion** at 3–4 weeks (ELISA) |
| Lagi 2011 [21][a,b] | Jan 2006- Dec 2010 | 12 hospitals in Tuscany, Italy | 2009 | **IgM positive**: Dengue IgG/IgM Combo Rapid Test (Cypress) or Dengue IgM & IgG Capture ELISA (Panbio) |
| Fonseca 2012 [22][a] | April-May 2011 | Ribeirao Preto, Brazil | 2009 | **NS1 positive**—acute sample **IgM positive**—acute sample |
| Nujum 2012 [23] | February 2011 to July 2011 | Primary and secondary care settings of Thiruvananthapuram district, Kerala state, India | 2009 | **RT-PCR positive** if fever <5 days **IgM antibody positive** (Standard Diagnostics) if fever >5 days–single sample only |
| Capeding 2013 [24] | Participant recruitment June-September 2010 Study conduction June 2010 to July 2011 | Indonesia (3 hospitals and 3 satellite centres), Malaysia (2 hospitals, 3 satellite clinics), Philippines (6 government health centres across 3 cities), Thailand (3 hospitals), Vietnam (1 hospital) | 1997 | **NS1 antigen positive in the acute sample**—*Platelia Dengue NS1Ag-ELISA (Bio-Rad)* **IgM antibody** in acute or convalescent sample—*Dengue Virus IgM Capture DxSelect ELISA kit (Focus Diagnostics)* **Four-fold rise in IgG antibody titres**—*Dengue Virus IgG Capture DxSelect ELISA kit (Focus Diagnostics)* |
| Daumas 2013 [25] | January 2005-June 2008 | Rio de Janeiro public hospital, Brazil– outpatient clinic for AFIs | 1997 | **NS1 antigen positive** (fever <5 days)—*Platelia Dengue NS1Ag-ELISA (Bio-Rad)* **RT-PCR positive** (fever <5 days)–*IPEC-FIOCRUZ laboratory* **IgM antibody** in the acute or convalescent sample–*MAC ELISA, PanBio* |
| | November 2007 to Jan 2008 | Emergency Department of a public general hospital, Brazil | | |
| Gutiérrez 2013 [16]– cohort study | August 2004 to December 2011 | Health Centre Sócrates Flores Vivas (HCSFV), Managua, Nicaragua | 1997 and 2009 | **RT-PCR positive** **Viral isolation** **IgM seroconversion**–*MAC-ELISA* **Four-fold rise in antibody titre**–*Inhibition ELISA* |
| Gutiérrez 2013 [16]– Hospital study | August 2005-January 2012 | Infectious Disease Ward of the Hospital Infantil Manuel de Jesús Rivera Hospital (HIMJR), Managua, Nicaragua | 1997 and 2009 | |
| Gan 2014 [26] | October 2011 – May 2012 | Communicable Disease Centre, Tan Tock Seng Hospital, Singapore | 1997 and 2009 | **Viral isolation**—*C6/36 cells* **RT-PCR positive** **NS1 antigen positive**—*Platelia Dengue NS1Ag-ELISA (Bio-Rad)* **IgG or IgM seroconversion**—*Panbio Dengue IgG Indirect or IgM Capture ELISAs (Alere Inc., Waltham, MA, USA)* **Four-fold rise in IgG titres**—*Panbio Dengue IgG Indirect ELISA (Alere Inc., Waltham, MA, USA)* |

*(Continued)*

**Table 3.** (Continued)

| Study | Data collection period | Location | Definition assessed | Reference standard(s) |
|---|---|---|---|---|
| Nujum 2014 [27] | Not specified | Outpatient departments and casualty of primary, secondary, and tertiary health care institutions of Thiruvananthapuram, Kerala, India | 2009 | **RT-PCR positive** if fever <5 days<br>**IgM antibody positive** (Standard Diagnostics) if fever >5 days–single sample only |
| Pitisuttithum 2015 [28][b] | October 2003 – June 2009 | Community-based, Rayong and Chonburi provinces, Thailand (RV144 trial participants) | 2009 | **Four-fold rise in antibody titres**—*hemagglutination inhibition assay, Clarke and Casals method* |
| Nealon 2016 [29] | 2011–2013 | Indonesia (3 centres)<br>Malaysia (2 centres)<br>Philippines (2 centres)<br>Thailand (2 centres)<br>Vietnam (2 centres) | 1997 | **NS1 antigen positive**—*ELISA*<br>**RT-PCR positive** |
| Seshan 2017 [30] | Not specified | Sri Ramachandra Medical College and Research Institute, Chennai, India | 2009 | **RT-PCR positive**<br>**NS1 antigen positive**—*Dengue Early ELISA (Panbio)*<br>**IgM antibody positive** (single sample)–*Dengue IgM Capture ELISA (PanBio)*<br>**IgG antibody positive** (single sample)–*Dengue IgG Indirect ELISA (PanBio)* |
| Caicedo 2019 [17]–Aedes Network[b] | Database 1: 2003–2011 | Multicentre cohort: Bucaramanga, Neiva, Cali, and Palmira; Colombia | 1997 and 2009 | **RT-PCR positive**<br>**NS1 antigen positive**–*Panbio ELISA*<br>**IgM serovonversion or four-fold rise**—*Panbio Dengue IgM Capture ELISA*<br>**Hemagglutination Inhibition titres ≥1: 2,560** |
| Caicedo 2019 [17]–National Public Health Surveillance[b] | Database 2: March-December 2012 | Cali, Colombia | 1997 and 2009 | **NS1 antigen positive**—*ELISA*<br>**RT-PCR positive**–*CDC, internal* |
| **STUDIES USING MODIFIED CRITERIA** | | | | |
| Peragallo 2003 [31] | February 15–28, 2000 | Italy (returning from East Timor) | Modified WHO 1997:<br>2–7 days fever<br>2+ of: headache, retrorbital pain, myalgia, arthralgia, cutaneous rash | **Hemagglutination inhibition titres ≥1:1,280 for DEN-2**<br>**Neutralisation test titres ≥1:20 for any DENV serotype** |
| Juárez 2005 [32] | April-May 2005 | Comas District, Lima | Modified WHO 1997:<br>2–7 days fever<br>2+ of: headache, retroocular pain, myalgia, arthralgia, and rash | **Virus isolation** if fever <5 days<br>**IgM ELISA** if fever >5 days |
| Low 2011 [33] | April 2005 – August 2010 | Community polyclinics in Singapore | WHO 1997 definition without tourniquet test<br>WHO 2009 definition but abdominal pain, mucosal bleeding and drowsiness were the only warning signs included | **RT-PCR positive**<br>**IgM seroconversion**–*ELISA (Panbio)* |
| Wieten 2012 [34][b] | 2006–2011 | Academic Medical Centre, Amsterdam | WHO 1997 without tourniquet test<br>WHO 2009 without tourniquet test | *Tests*: Dengue Duo rapid strip test (Panbio)<br>Dengue Duo IgM/IgG capture ELISA (Panbio)–**first 15 months only**<br>If two samples: **IgM seroconversion**<br>**Four-fold increase in IgG titres**<br>If one sample: **Positive IgM or IgG** >5 days after symptom onset<br>**Positive/borderline IgM or IgG** <5 days after symptom onset |

*(Continued)*

**Table 3.** (Continued)

| Study | Data collection period | Location | Definition assessed | Reference standard(s) |
|---|---|---|---|---|
| Ridde 2016 [35] | December 2013 – January 2014 | Six primary healthcare centres in Ouagadougou, Burkina Faso | WHO 2009 limited to the following: nausea/vomiting, 'aches and pains', rash, tourniquet test, abdominal pain, lethargy/sleepiness, convulsions, and mucous membrane bleeding. | **IgM and/or IgG positive**—*Dengue Duo rapid test (Standard Diagnostics)* **RT-PCR positive** if positive RDT, and every 10th subject with negative RDT |
| Bodinayake 2018 [36] | June 2012 –May 2013 | Tertiary care hospital (Teaching Hospital Karapitiya) in Southern Province, Sri Lanka | WHO 2009 without tourniquet test | **IgG seroconversion alone with positive IgM or IgM seroconversion**–*in-house ELISA* **PCR positive and viral isolation or alternative target PCR** **PCR positive and/or viral isolation with a positive convalescent IgM** |

ELISA: Enzyme-linked immunosorbent assay. (RT-)PCR: (Reverse transcriptase) polymerase chain reaction. NS1: Non-structural protein 1.

a conference abstract.

b retrospective study.

## Symptom associations

In studies that assessed the association of clinical/laboratory variables with a confirmed dengue infection, leukopenia [17,18,21,25,27,28,33,36] and thrombocytopenia [17,21,23,25,27,28,33,36] were the most frequently associated, consistent with previous studies [37,38]. Other notable associations include rash [16,17,25,32,33] and haemorrhagic manifestations (including the tourniquet test) [16–19], reported as the most specific features, although lacking sensitivity. Two studies found significant associations with taste disorders [25,33], a symptom not in either definition.

## Effects of age on diagnostic accuracy

The sensitivity of both definitions was halved in patients under 4 years presenting in the community. The reduction was less marked for hospital presentations: approximately 10% for the 1997 definition and 2% for the 2009 definition [16]. This could be due to children's inability to report symptoms such as retro-orbital pain, myalgia, and arthralgia. In theory, the 2009 definition (which combines them as 'aches and pains') should overcome this but does not appear to do so in practice. In both community and hospital settings, this fall in sensitivity was accompanied by an increase in specificity, again less marked in hospital settings [16].

At the other extreme of age, the frequency of many symptoms associated with dengue fever, such as retro-orbital pain and mucosal bleeding, decreased with increasing age, particularly over 56 years. This led to decreasing sensitivity of both definitions in older adults [33].

Dengue may present differently in adults and children. Children (but not adults) with dengue were more likely to have sore throat, fatigue, oliguria, and elevated haematocrit and transaminases compared to children with other febrile illnesses. Conversely, adults were more likely to have joint pain [36].

## Discussion

In this review, we have pooled evidence from multiple regions assessing the accuracy of the 1997 and 2009 WHO clinical definitions for diagnosing dengue fever. We have shown that both definitions have high sensitivity (93%) but poor specificity (29% and 31%). No

**Table 4. Patient flow through studies included in the systematic review.**

| Study | Inclusion criteria | Exclusion criteria | Patients in final analysis/total number of patients | Reasons for exclusion (if applicable) |
|---|---|---|---|---|
| Sawasdivorn 2001 [18] | Ages not specified<br>Provisional diagnosis of dengue infection or suspected dengue infection.<br>Parental consent. | Not specified | 176/176 | N/A |
| Martinez 2005 [19] | Age >12 years<br><96h of fever<br>Informed consent<br>Diagnostic impression of dengue or unspecified viral infection. | Clinical evidence of another infectious process that partially or totally explains the current disease.<br>Diabetes, AIDS, cirrhosis, rheumatological or malignant disease, heart or kidney failure, or history of corticosteroid use.<br>Residence in a rural area or with difficult access for monitoring. | 190/190 | N/A |
| Gan 2011 [20] | Suspected dengue cases | Not specified | 162/205 | Lack of paired sera or elevated IgM/IgG without seroconversion—43 |
| Lagi 2011 [21] | Not specified | Missing clinical or lab data | 109/109 | N/A |
| Fonseca 2012 [22] | Not specified<br>Convenience sample | Not specified | 1490/1490 | N/A |
| Nujum 2012 [23] | Ages not specified<br>2–7 days of acute febrile illness<br>Household surveys around confirmed dengue cases and corresponding primary/secondary care settings | Definite diagnosis<br>Informed consent not given | 254/254 | N/A |
| Capeding 2013 [24] | Age 2–14 years on the day of enrolment<br>>2 days fever<br>Able to attend scheduled visits/comply with study procedures | History of chronic illness/immunodeficiency<br>Pandemic influenza vaccination in 2 weeks before/after enrolment<br>Any other vaccination in 4 weeks before/after enrolment | 358/374 febrile episodes<br>1487/1500 patients | Voluntary withdrawal– 12<br>Noncompliant with protocol -1<br>Missing lab diagnosis—42 |
| Daumas 2013 [25] | Age >12 years<br>≤3 days fever<br>**For emergency department study:** first 8 patients, once a week | Severe illness (i.e. with altered consciousness, signs of shock or severely dehydrated) in need of emergency care<br>Evident or suspected focus on clinical examination (e.g. tonsillitis, pyelonephritis) | 142/182 | Indeterminate test result—40 |
| Gutiérrez 2013 [14]–cohort study | Age 2–15 years<br>≤6 days fever without apparent origin<br>Part of Paediatric Dengue Cohort Study | Not specified | 3407/3617 | Missing confirmed lab result—210 |
| Gutiérrez 2013 [16]–Hospital study | Age 6 months– 14 years<br><7 days fever<br>Inpatients/outpatients<br>1+ of: headache, arthralgia, myalgia, retro-orbital pain, positive tourniquet test, petechiae or other signs of bleeding. | Defined focus other than dengue<br>Weight <8 kg<br>Age >6y with signs of altered consciousness at the time of recruitment and thus unable to provide assent | 1160/1210 | Indeterminate lab results—50 |
| Gan 2014 [26] | Age ≥18 years<br>Fever >37.5°C | Pregnancy<br>Alternative syndromic diagnosis for febrile illness | 197/246 | Point-of-care test not performed– 2<br>Inconclusive reference standard—47 |
| Nujum 2014 [27] | Age >5 years<br>2–7 days fever | Already diagnosed as dengue<br>Not giving informed consent<br>Definitive diagnosis/focus of infection | 851/939 | Consent not given—89 |

(*Continued*)

**Table 4.** (Continued)

| Study | Inclusion criteria | Exclusion criteria | Patients in final analysis/total number of patients | Reasons for exclusion (if applicable) |
|---|---|---|---|---|
| Pitisuttithum 2015 [28] | All patients with dengue recorded as an adverse event/severe adverse event.<br>All individuals without a dengue diagnosis and a severe adverse event using a preferred term that corresponded to the system organ class "Infections and infestations" or idiopathic fever (pyrexia) occurring between June and September (missed dengue cases) | Missing acute/convalescent blood sample<br>Diagnoses with a known cause<br>Individuals whose samples were identified as controls for the immune correlates study | 121/124 dengue events<br>72/77 non-dengue SAEs | Missing serology specimens—8 |
| Nealon 2016 [29] | 2–14 years<br>CYD14 vaccine trial participant<br>Fever ≥ 38°C on ≥ 2 consecutive days | Not specified | 3099/3099 febrile episodes | N/A |
| Seshan 2017 [30] | Acute undifferentiated febrile illness | Not specified | 150/150 | N/A |
| Caicedo 2019 [17]–Aedes Network | Ages not specified<br><96h fever of unknown origin/suspected dengue<br>Dengue diagnosis in the first 24h of hospitalisation | Concomitant diseases | 987/987 | N/A |
| Caicedo 2019 [17]–National Public Health Surveillance | National Public Health Surveillance system notification sheets | Not specified | 461/461 | N/A |
| **STUDIES USING MODIFIED CRITERIA** | | | | |
| Peragallo 2003 [31] | Troops returning to Italy after 3-month period of duty in East Timor | Not specified | 241/280 | Not specified |
| Juárez 2005 [32] | Clinical picture compatible with an infectious process<br>Received National Institute of Health diagnostic tests<br>Patient recorded in PHLIS database | Incomplete records<br>Diagnostic test not specified<br>Yellow fever vaccination <10d before symptom onset | 315/479 | Not specified |
| Low 2011 [33] | Age ≥18 years<br><72h of fever >37.5°C | Not specified | 2129 | Not specified |
| Wieten 2012 [34] | All patients who were serologically tested for dengue at the AMC between 2006 and 2011 | Treated outside the AMC<br>Double tests<br>Missing files<br>Onset of symptoms >6 months before consultation<br>No travel history | 409/1124 | Treated outside the AMC—200<br>Double tests– 285<br>Missing files– 31<br>Onset of symptoms >6 months before consultation– 14<br>No travel history– 13<br>Indeterminate results– 172 |
| Ridde 2016 [35] | Fever >38°C at survey or during the previous week<br>Negative rapid diagnostic test for malaria | Not specified | 379/379 | N/A |
| Bodinayake 2018 [36] | Age ≥1 year<br>Admitted to a medical or paediatric non-surgical ward with documented fever (>38°C) at presentation or within 48 hours of hospital admission | Hospitalized for >48 hours<br>Hospitalized/underwent surgery in the previous 7 days<br>Unable/unwilling to give consent | 838/877 | Inconclusive results—39 |

modification improved accuracy. This makes the definitions useful rule-out criteria but unreliable as the basis for diagnosis, which is concerning given they are often used as such [8–10].

Clinical presentation varied with age, with diagnostic accuracy suffering at the extremes of age. As the average age of dengue cases increases, case definitions developed from paediatric studies [39] will no longer be sufficient. Overall, our findings highlight the need for an urgent reassessment of these guidelines.

## SELECTION  INDEX  REFERENCE  FLOW

| | SELECTION | INDEX | REFERENCE | FLOW |
|---|---|---|---|---|
| Sawasdivorn 2001 | + | + | ? | +* |
| Martinez 2005 | + | + | + | +* |
| Gan 2011 | ? | ? | ? | +* |
| Lagi 2011 | ? | ? | - | ? |
| Fonseca 2012 | - | + | - | +* |
| Nujum 2012 | + | + | - | +* |
| Capeding 2013 | + | + | - | +* |
| Daumas 2013 | - | + | - | +* |
| Gutiérrez 2013 - CS | + | + | + | + |
| Gutiérrez 2013 - HS | + | ? | + | +* |
| Gan 2014 | ? | + | + | + |
| Nujum 2014 | + | ? | - | +* |
| Pitisuttithum 2015 | - | + | - | - |
| Nealon 2016 | + | + | + | +* |
| Seshan 2017 | ? | + | - | +* |
| Caicedo 2019 - AN | + | + | + | +* |
| Caicedo 2019 - PHS | + | + | + | +* |
| Peragallo 2003 | + | ? | - | - |
| Juarez 2005 | + | - | - | +* |
| Low 2011 | + | + | + | +* |
| Wieten 2012 | - | - | - | ? |
| Ridde 2016 | + | - | - | +* |
| Bodinayake 2018 | + | + | + | +* |

MODIFIED (bracket spanning Peragallo 2003 through Bodinayake 2018)

**Fig 2. Risk of bias analysis for included studies assessing the diagnostic accuracy of clinical criteria for dengue fever.** + (green): Low risk; +* (green): Withdrawals/indeterminate results not mentioned, assumed none and thus low risk;? (amber): unclear risk;—(red): high risk; CS: cohort study; HS: Hospital study; AN: Aedes Network study; PHS: Public Health Surveillance study. Modified refers to modifications from the 1997 or 2009 WHO criteria. The assessment was carried out by two independent reviewers (NR/SL) using a modified version of QUADAS-2 (S2 Table). A study was deemed to be at high risk of bias if any domain was at high risk of bias.

## 1997 DEFINITION

**Fig 3. Forest plot for sensitivity and specificity of the WHO 1997 case definition for probable dengue.** Meta-analysis carried out in Stata/IC 14 using the MIDAS statistical package. 95% confidence intervals given in parentheses. HS: hospital study; CS: cohort study; AN: Aedes Network Study; PHS: Public Health Surveillance Network Study. a: Calculated specificity differs from the reported value (0.2121); b: Calculated sensitivity differs from the reported value (0.98).

Two outliers (both assessing the 1997 definition) displayed the reverse pattern with high specificity and low sensitivity [24,29]. Interestingly, they both employed an active surveillance study design that monitored a community cohort for febrile illness. Given the high expansion factors associated with dengue fever [5], healthcare systems have a low sensitivity for detecting dengue cases, and thus patients presenting to health services (the majority of included studies) may not be representative of dengue cases overall, which could explain why the case definitions perform so differently in a more representative population. However, the only other included study that used a prospective fever design found a low specificity and high sensitivity [16], so caution is needed in interpreting the conclusions from these two outliers as they were both performed consecutively in the same centres and another confounding factor may be contributing. In addition, while not representative of all dengue cases, patients presenting to healthcare services are more representative of what frontline clinicians see daily. Further research is therefore warranted to better understand this differential performance of the case definitions, as it may have opposing implications for public health surveillance and clinical practice.

Seshan et al. and Lagi et al. also found a better specificity than expected (63%), but both used single samples for IgM/IgG serology as the reference standard, which would lead to more

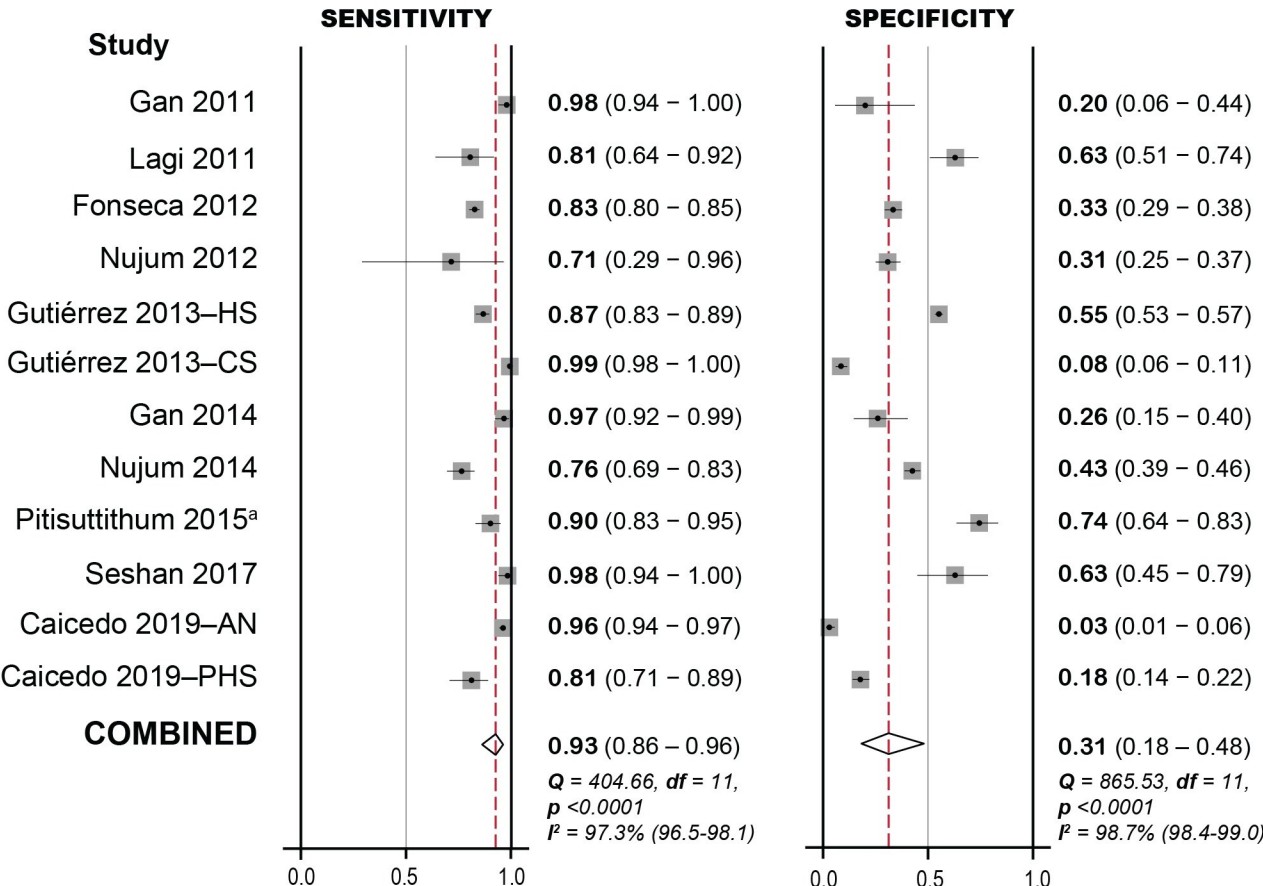

**Fig 4. Forest plot for sensitivity and specificity of the WHO 2009 case definition for probable dengue.** Meta-analysis carried out in Stata/IC 14 using the MIDAS statistical package. 95% confidence intervals given in parentheses. HS: hospital study; CS: cohort study; AN: Aedes Network Study; PHS: Public Health Surveillance Network Study. a: Calculated sensitivity differs from the reported value (0.909).

**Table 5. Sensitivity and specificity from studies that modified WHO clinical criteria for dengue diagnosis.**

| Study | Sensitivity (95% CI) | Specificity (95% CI) |
|---|---|---|
| Peragallo 2003 –modified 1997 | 0.73 (0.50–0.88) ↓ | 0.78 (0.65–0.87) ↑ |
| Juárez 2005 –modified 1997 | 0.81 (0.74–0.86) ↓ | 0.28 (0.21–0.36) ↓ |
| Low 2011 –modified 1997 | 0.93 (0.89–0.96) | 0.32 (0.30–0.34) ↑ |
| Low 2011 –modified 2009 | 0.95 (0.91–0.97) ↑ | 0.23 (0.21–0.25) ↓ |
| Wieten 2012 –modified 1997 | 0.84 (0.77–0.90) ↓ | 0.43 (0.37–0.49) ↑ |
| Wieten 2012 –modified 2009 | 0.46 (0.39–0.52) ↓ | 0.14 (0.10–0.21) ↓ |
| Bodinayake 2018 –modified 2009 | 0.76 (0.71–0.80) ↓ | 0.65 (0.60–0.69) ↑ |
| Ridde 2016 –modified 2009 | 1.0 (0.89–1.0) ↑ | 0.69 (0.64–0.74) ↑ |

Green shading indicates better performance than the corresponding definition's summary estimate (i.e. better than unmodified criteria). Red shading indicates the opposite.

false positives in the reference standard, thus overestimating specificity of the index test (clinical diagnosis) [21,30]. Pitisuttithum et al. also found a higher specificity (74%), which could be due to their case-control study design which may not capture all febrile presentations like the other studies [28].

## Surveillance implications of inaccurate clinical diagnosis

While underreporting remains a major issue for dengue [1,5], given the low specificity of the clinical definitions it is highly likely that non-dengue viral illness is also being misreported as dengue. This makes it difficult to assess the burden and spread of dengue across regions, particularly during outbreaks. While dengue is the most common cause of acute febrile illness in Southeast Asia [38] and Latin America [40], other causative agents include the arboviruses Chikungunya [24,37,38,40] and Zika [40]; respiratory viruses (e.g. influenza) [24,33,37]; and bacteria such as rickettsia and leptospirosis [38,40]. The co-circulation of multiple pathogens causing similar clinical pictures is uncontroversial, and, as evidenced by our findings, not what the clinical definitions were developed to handle.

This poses an issue to public health policy, surveillance, and response measures. A large number of (false-positive) dengue referrals to tertiary care may overwhelm healthcare systems, particularly during 'outbreaks' [23]. Chikungunya and Zika share the same vector and thus may be amenable to the same control measures. However, the inability to determine which *Aedes*-borne virus is responsible for a particular case cluster makes it difficult to assess the introduction of novel viruses to an area and trigger early responses.

With the licensing of the new dengue vaccine, governments need to prioritise areas where vaccine introduction will have the most impact and thereafter measure its efficacy. This is made exceedingly difficult if they cannot ascertain which pathogen is primarily responsible for a region's disease burden. In resource-limited settings, this uncertainty, at the level of both the individual patient and the surveillance system, carries significant opportunity cost for which treatments, control measures, and vaccines to prioritise.

## Impact of COVID-19

These issues have only increased in importance during the COVID-19 pandemic. There are case reports of COVID-19 being misdiagnosed as dengue [41], including due to an atypical presentation with a rash [42] (a relatively specific feature of dengue). The studies included in this systematic review predate the COVID-19 pandemic, making it difficult to draw direct conclusions on the effect COVID-19 has on clinical dengue diagnosis. However, the overlap in syndromic definitions [43] and the prevalence of cough and respiratory symptoms in over a third of dengue patients [25,36,37] makes it likely that dengue is also being misdiagnosed as COVID-19 [4]. In particular, the association of taste disorders (a cardinal symptom of COVID-19 [43]) with dengue [25,33] warrants further investigation.

Misclassification of COVID-19 as dengue and vice versa has a profound impact on public health responses due to the very different control measures. Control of dengue relies on control of mosquito vectors or reducing human-vector contact. This generally relies heavily on visits to households, workplaces, schools and other mosquito breeding sites for environmental management, and application of chemical and/or biological measures [12]. This is in stark opposition to COVID-19 control which relies on lockdown measures including restrictions on travel and social interaction. Dengue control has thus been negatively affected during the pandemic [2]. Abandoned buildings (e.g. due to school closures) and lack of maintenance of public spaces can contribute to increases in mosquito populations [3]. As countries report rises in both dengue and COVID transmission [2,3], governments need accurate transmission figures

(and hence clinician access to rapid and accurate diagnostics) to prioritise region-specific control measures [4].

## Implications for diagnostic testing

While mandating confirmatory testing may increase the specificity of dengue diagnosis, despite recent developments in diagnostics (including the highly-accurate NS1 antigen detection tests) [6], rapid and accurate laboratory confirmation remains inaccessible in most dengue-endemic regions. Furthermore, cheaper tests such as IgM and IgG serology are likely to become less useful as dengue vaccination programs are rolled out; their already (relatively) low specificity has been demonstrated to fall in vaccinated individuals [44].

Once again, this is exacerbated by COVID-19, with case reports of false-positive dengue serology in COVID-19 [41] and a study in a dengue non-endemic area showing 22% false dengue positivity amongst COVID patients (albeit in a small sample) [45]. These findings suggest that the need for better clinical guidance (or cheaper diagnostics) is likely to become increasingly urgent as dengue serology, the most common and accessible laboratory test, becomes less informative.

## Possible improvements to the clinical case definitions

One possibility is to use the absence of features more strongly associated with other aetiologies as supporting criteria [17]. For example, the absence of cough, lung crackles [36], and backache [23] were found to be significantly associated with dengue. However, while less common, they are still present in a significant proportion of dengue cases, and therefore their absence can only be a supporting sign.

Another possibility could be prioritising symptoms within case definitions, perhaps by splitting into 'major' and 'minor' criteria, so that symptoms more strongly associated with dengue, such as leukopenia or thrombocytopenia, carry more weight in making the diagnosis. As this was not the goal of this systematic review, further research is needed to better identify these symptom constellations.

Similar to guidance for laboratory testing [12], the case definition could be modified so that symptom criteria vary depending on timing within the illness course, where symptom associations are known to differ [37]. For example, platelet count, while reduced in dengue patients, may be normal at first, making thrombocytopenia more informative later in the illness [33,37]. Another study found that headache, myalgia, and retro-orbital pain were more sensitive earlier on, whilst rash was the opposite [32]. Modifying the definition at different timepoints may therefore improve accuracy, although current findings demonstrate inconsistencies and further research and/or systematic review is necessary.

Test positivity also varies over time with different windows for detectable PCR, NS1, IgM and IgG [12]. Some studies took this into account in deciding which reference standard to use, as outlined in Table 3. Differences in test timing may contribute to some of the variability in sensitivity/specificity found between studies. However, as most included studies did not state whether the choice of reference standard varied depending on illness duration, the impact of this could not be adequately assessed. This is a potential topic for future research.

Nonetheless, any clinical definition will remain imperfect given the variable and nonspecific presentation of dengue. Alternatively, modified case definitions could guide the allocation of limited testing resources rather than diagnosis. Specificity increases when more criteria are required (e.g. 5 instead of 2) [16,19,23], increasing diagnostic certainty. Thus, as dengue can effectively be ruled out in patients not fulfilling the criteria (due to the high sensitivity), and is highly likely in those with multiple symptoms, laboratory confirmation can be reserved for

those patients with only 2 or 3 symptoms where uncertainty is greatest and testing will be most informative [23].

Finally, local guidelines or electronic decision support tools could incorporate epidemiological information about circulating pathogens to prioritise symptoms and signs that would be most discriminatory for the region's differentials [4,46]. As clinician diagnosis at both admission and discharge was more specific (but less sensitive) than WHO criteria [36], this could already be considered by experienced clinicians and is a potential avenue for further research.

### Strengths and limitations

Our study conformed to PRISMA guidelines (S7 Table) and was methodologically robust. By using two independent reviewers, researcher bias was mitigated at every stage of analysis. By searching multiple databases (including grey literature) and carrying out a thorough citation analysis we believe we have captured most, if not all, the available literature on the diagnostic accuracy of dengue case definitions. The inclusion of studies from multiple regions increases the generalizability of our findings. Only one eligible study from Africa could be found. This may be due to a lack of dengue or a lack of dengue research in this region, which could itself be a result of underrecognition and underdiagnosis.

The main limitation was the significant heterogeneity (in methods and results) of included studies and the high risk of bias. This is likely due to the use of different reference standards between studies. As diagnostic accuracy varies between and within confirmatory tests [6], and no test is perfect, this would introduce significant bias to results (especially when IgM or IgG serology alone were used for confirmation). Furthermore, the different spectra of illness presenting in different healthcare settings and age groups may also contribute to heterogeneity in clinical case definition performance.

However, except for two outliers, studies across different regions, healthcare settings, and patient ages demonstrated relatively high sensitivity and poor specificity. While the summary values should be used with caution, the need for urgent improvement in dengue diagnostic guidance and reporting practice is clear.

Nonetheless, it is worth noting that, unlike the studies included in this systematic review, frontline clinicians may not apply WHO criteria strictly without also considering contextual epidemiology (such as a local outbreak). The effect of this on the accuracy of clinical diagnosis (and subsequent reporting of global cases) remains unclear. It may improve through correctly dismissing cases that fulfil the WHO criteria when other circulating pathogens are more common (out of 'dengue season') but may also lead to self-fulfilling prophecies of dengue outbreaks due to the nonspecific nature of the case definitions. This overdiagnosis could be offset by clinicians being too busy during outbreaks to report all cases, hence why studies may not find evidence of over/underreporting during outbreaks. Further research would be helpful in understanding the impact of outbreaks on reporting rates in light of limited access to testing and nonspecific case definitions.

### Conclusion

This review has demonstrated the poor diagnostic accuracy of the clinical definitions for dengue in the absence of confirmatory testing. This has real-world costs both for treating clinicians and for surveillance systems, magnified by COVID-19. As fragile healthcare systems prepare to cope with the possibility of double epidemics, further research into improved clinical guidance, access to diagnostic testing, and accurate quantification of dengue burden and transmission will be essential.

## Supporting information

**S1 File. Description of methods.**
(DOCX)

**S1 Table. Search strings used in systematic review.**
(DOCX)

**S2 Table. Modified QUADAS-2 quality assessment tool.**
(DOCX)

**S3 Table. Anticipated effect of study bias on sensitivity/specificity estimates.**
(DOCX)

**S4 Table. Data from studies looking at WHO 1997 definition.**
(DOCX)

**S5 Table. Data from studies looking at WHO 2009 definition.**
(DOCX)

**S6 Table. Data from studies using modified WHO criteria.**
(DOCX)

**S7 Table. PRISMA checklist for systematic review.**
(DOCX)

**S1 Fig. Deeks' Funnel Plot analysis of publication bias– 1997 definition.** 1, Sawasdivorn 2001 [18]; 2, Martinez 2005 [19]; 3, Gan 2011 [20]; 4, Capeding 2013 [24]; 5, Daumas 2013 [25]; 6, Gutiérrez 2013 –cohort study [16]; 7, Gutiérrez 2013 –hospital study [16]; 8, Gan 2014 [26]; 9, Nealon 2016 [29]; 10, Caicedo 2019 –Aedes Network Study [17]; 11, Caicedo 2019 – Public Health Surveillance Network study [17].
(TIF)

**S2 Fig. Deeks' Funnel Plot analysis of publication bias– 2009 definition.** 1, Gan 2011 [20]; 2, Lagi 2011 [21]; 3, Fonseca 2012 [22]; 4, Nujum 2012 [23]; 5, Gutiérrez 2013 –cohort study [16]; 6, Gutiérrez 2013 –hospital study [16]; 7, Gan 2014 [26]; 8, Nujum 2014 [27]; 9, Pitisut-tithum 2015 [28]; 10, Seshan 2017 [30]; 11, Caicedo 2019 –Aedes Network Study [17]; 12, Cai-cedo 2019 –Public Health Surveillance Network study [17].
(TIF)

## Author Contributions

**Conceptualization:** Nader Raafat, Stuart D. Blacksell, Richard James Maude.

**Data curation:** Nader Raafat, Shanghavie Loganathan.

**Formal analysis:** Nader Raafat, Mavuto Mukaka.

**Funding acquisition:** Richard James Maude.

**Investigation:** Nader Raafat, Shanghavie Loganathan, Stuart D. Blacksell.

**Methodology:** Nader Raafat.

**Software:** Mavuto Mukaka.

**Supervision:** Richard James Maude.

**Validation:** Richard James Maude.

**Visualization:** Nader Raafat.

**Writing – original draft:** Nader Raafat.

**Writing – review & editing:** Shanghavie Loganathan, Mavuto Mukaka, Stuart D. Blacksell, Richard James Maude.

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
