## [Decision Letter · Decision Letter 0]

5 Jan 2021

Dear Prof. Maude,

Thank you very much for submitting your manuscript "Diagnostic accuracy of the WHO clinical definitions for dengue and implications for surveillance: a systematic review and meta-analysis" for consideration at PLOS Neglected Tropical Diseases. As with all papers reviewed by the journal, your manuscript was reviewed by members of the editorial board and by several independent reviewers. The reviewers appreciated the attention to an important topic. Based on the reviews, we are likely to accept this manuscript for publication, providing that you modify the manuscript according to the review recommendations. 

Sincerely,

Rhoel Ramos Dinglasan

Associate Editor

Robert Reiner

Deputy Editor

Reviewer's Responses to Questions

**Key Review Criteria Required for Acceptance?**

**Methods**

-Are the objectives of the study clearly articulated with a clear testable hypothesis stated?

-Is the study design appropriate to address the stated objectives?

-Is the population clearly described and appropriate for the hypothesis being tested?

-Is the sample size sufficient to ensure adequate power to address the hypothesis being tested?

-Were correct statistical analysis used to support conclusions?

-Are there concerns about ethical or regulatory requirements being met?

Reviewer #1: I think the authors have performed a very thorough search for clinical papers and therefore this review gives a good picture of the field.

My main concern is that the outcome isn’t really different from what we already knew. I am wondering whether this is due to the enormous heterogeneity in studies included. The risk is that the outcome of the statistical analysis deviates to the mean. The authors indicate this in their discussion and give options about how to improve the classification. I am curious if they have identified studies in which the case classification performed better than expected and why this happened.

The objectives of the study are clear and the study design appropriate.

In some studies the inclusion criteria or the age categories are not clear. I wonder how the authors handled that.

The sample size is sufficient.

The statistical analysis was correct.

Reviewer #2: The methods are detailed and generally clear. However, a few refinements would be useful. In assessing the risk of bias, it would be helpful for the authors to indicate the anticipated impact of the suspected bias. This could use a nomenclature such as the following, reporting on both sensitivity and specificity (with a code such as 0 if there is no bias hypothesized for that measure):

SEN+ (the sensitivity calculated from the study data is likely to be too high compared to an unbiased study), 

SEN- (the sensitivity calculated from the study data is likely to be too low compared to an unbiased study), 

SPEC+ (the specificity calculated from the study data is likely to be too high compared to an unbiased study), 

SPEC- (the specificity calculated from the study data is likely to be too low compared to an unbiased study).

In a separate supplemental table, the authors should explain the rationale for their choices. This process would be a useful contribution to the field in indicating whether the results confirm or contradict the hypothesized bias.

**Results**

-Does the analysis presented match the analysis plan?

-Are the results clearly and completely presented?

-Are the figures (Tables, Images) of sufficient quality for clarity?

Reviewer #1: The analysis presented matches the analysis plan. The results are clearly presented and the figures are of sufficient quality.

Reviewer #2: The results are generally well presented.

**Conclusions**

-Are the conclusions supported by the data presented?

-Are the limitations of analysis clearly described?

-Do the authors discuss how these data can be helpful to advance our understanding of the topic under study?

-Is public health relevance addressed?

Reviewer #1: The conclusions are supported by the data presented.

According to my opinion epidemiological data play an important role in clinical decision making. If the MD knows that a certain region is targeted by a DENV outbreak he/she will make this diagnosis sooner. Therefore I think the authors should acknowledge that this also plays an important role in the global reports of DENV cases and not only the clinical case classification. This may also help in improving the case classification. 

The authors write a lot about how the low specificity of the case classification could be harmful in the COVID-19 era. In my opnion they should be cautious with this statements, because the studies included date back from before this.

The authors discuss extensively how the data can be helpful to understand the topic and public health relevance is also extensively discussed.

Reviewer #2: The authors nicely note that diagnostic criteria are helpful both to guide clinical management and for informing public health policy. Thus, their guidance on lab testing for borderline cases is useful. The value of the study would be further enhanced if the authors could use their data to suggest further refinements to the classification of dengue symptoms to increase specificity. They could consider terms such as dengue with "some," "substantial," or "overwhelming" supporting evidence. These terms would presumably translate into increasingly higher specificities.

Another concern that it would be useful to address is the different time windows of the NS1 and IGg and IGm tests. If the NS1 test is taken too long after dengue infection or the IGG or IGM tests are taken too late, they will show up negative. The authors should discuss this and the extent to which the literature can address the problem.

A broader concern deserving discussion is reconciling the high expansion factors (e.g. 7-fold) which implies a sensitivity for the health care system of only about 14%. This discrepancy implies that the populations used for this meta-analysis are not very representative of dengue cases overall. Prospective fever studies in a given location would seem especially useful for public health purposes. The authors may wish to discuss which, if any, of the studies had that design and whether and how the results differed from the overall studies that they reviewed.

Finally, the authors may wish to discuss the outliers, such as the study by Capeding from Indonesia. Why were the findings so different from the others and what are the implications.

**Editorial and Data Presentation Modifications?**

Reviewer #1: Introduction:

In the introduction the authors highlight the fact that especially with the COVID-19 pandemic it is important to distinguish DENV-infection from COVID-19. From the text it is not really clear why this is important, but I can imagine it is especially for the isolation measurements. I am wondering in areas where testing on DENV is not common how the testing procedures on COVID-19 are.

Line 67: ‘not unlike the ongoing COVID-19 pandemic’: I don’t understand the meaning of this line. Asymptomatic spread also contributes to the COVID-19 pandemic.

Line 73: ‘is’ should be ‘are’

Discussion

304: I think with using laboratory markers it is not always difficult to distinguish bacterial infection from viral infection. Moreover, exposure plays an important role as well.

One limitation of this study is that report from the African region are missing, although I think there aren’t a lot.

Reviewer #2: Presentation is mostly clear. However, I would suggest that the authors replace the statement in lines 59-60 "as we discuss in this article." It would be helpful to summarize their recommendation in a phase, such as testing the borderline clinical diagnoses.

**Summary and General Comments**

Reviewer #1: none

Reviewer #2: The study is well implemented and clearly presented. However, its contribution to clinical and public health practice would be strengthened, I believe, by adding to the discussion and conclusion along the lines suggested.

PLOS authors have the option to publish the peer review history of their article (what does this mean?). If published, this will include your full peer review and any attached files.

Reviewer #1: Yes: Cornelia van de Weg

Reviewer #2: Yes: Donald S Shepard
---

## [Decision Letter · Decision Letter 1]

2 Apr 2021

Dear Prof. Maude,

We are pleased to inform you that your manuscript 'Diagnostic accuracy of the WHO clinical definitions for dengue and implications for surveillance: a systematic review and meta-analysis' has been provisionally accepted for publication in PLOS Neglected Tropical Diseases.

Best regards,

Rhoel Ramos Dinglasan

Associate Editor

Robert Reiner

Deputy Editor

Reviewer's Responses to Questions

**Key Review Criteria Required for Acceptance?**

**Methods**

-Are the objectives of the study clearly articulated with a clear testable hypothesis stated?

-Is the study design appropriate to address the stated objectives?

-Is the population clearly described and appropriate for the hypothesis being tested?

-Is the sample size sufficient to ensure adequate power to address the hypothesis being tested?

-Were correct statistical analysis used to support conclusions?

-Are there concerns about ethical or regulatory requirements being met?

Reviewer #1: Yes

**Results**

-Does the analysis presented match the analysis plan?

-Are the results clearly and completely presented?

-Are the figures (Tables, Images) of sufficient quality for clarity?

Reviewer #1: Yes

**Conclusions**

-Are the conclusions supported by the data presented?

-Are the limitations of analysis clearly described?

-Do the authors discuss how these data can be helpful to advance our understanding of the topic under study?

-Is public health relevance addressed?

Reviewer #1: Yes

**Editorial and Data Presentation Modifications?**

Reviewer #1: no more suggestions

**Summary and General Comments**

Reviewer #1: In my opinion the authors addressed all issues from us reviewers very accurately

PLOS authors have the option to publish the peer review history of their article (what does this mean?). If published, this will include your full peer review and any attached files.

Reviewer #1: **Yes: **Cornelia van de Weg

---

## [Editor Report · Acceptance letter]

20 Apr 2021

Dear Prof. Maude,

We are delighted to inform you that your manuscript, "Diagnostic accuracy of the WHO clinical definitions for dengue and implications for surveillance: a systematic review and meta-analysis," has been formally accepted for publication in PLOS Neglected Tropical Diseases.

Best regards,

Shaden Kamhawi

co-Editor-in-Chief

Paul Brindley

co-Editor-in-Chief
